# Learning From Simplicial Data Based on Random Walks and 1D Convolutions

**Florian Frantzen**
Department of Computer Science
RWTH Aachen University, Germany
`florian.frantzen@cs.rwth-aachen.de`

**Michael T. Schaub**
Department of Computer Science
RWTH Aachen University, Germany
`schaub@cs.rwth-aachen.de`

## Abstract

Triggered by limitations of graph-based deep learning methods in terms of computational expressivity and model flexibility, recent years have seen a surge of interest in computational models that operate on higher-order topological domains such as hypergraphs and simplicial complexes. While the increased expressivity of these models can indeed lead to a better classification performance and a more faithful representation of the underlying system, the computational cost of these higher-order models can increase dramatically. To this end, we here explore a simplicial complex neural network learning architecture based on random walks and fast 1D convolutions (SCRaWl), in which we can adjust the increase in computational cost by varying the length and number of random walks considered while accounting for higher-order relationships. Importantly, due to the random walk-based design, the expressivity of the proposed architecture is provably incomparable to that of existing message-passing simplicial neural networks. We empirically evaluate SCRaWl on real-world datasets and show that it outperforms other simplicial neural networks.

## 1 Introduction

In recent years, there has been a strong interest in extending graph-based learning methods, in particular graph neural networks, to higher-order domains such as hypergraphs and simplicial complexes. These higher-order abstractions offer a more comprehensive representation of relationships among entities than traditional graph-based approaches, which are based on pairwise interactions. Accordingly, it has been shown that we can achieve performance gains in various learning and prediction tasks for complex systems by employing such higher-order models (Benson et al., 2018; Schaub et al., 2021; Battiston & Petri, 2022). For instance, in the context of graph classification, it is well known that standard message-passing graph neural networks with anonymous inputs are limited in their expressiveness by the one-dimensional Weisfeiler-Leman graph isomorphism test (Morris et al., 2019; Xu et al., 2019; Morris et al., 2023) and thus cannot distinguish certain non-isomorphic graphs. A concrete example is that such graph neural networks cannot differentiate between a cycle of length 6 vs. two non-connected triangles. In contrast, higher-order extensions of graph-neural networks have greater expressiveness and naturally enable us to distinguish such graphs.

However, due to the combinatorial increase in the possible interactions, there is a significant cost in terms of increased memory and compute time associated with the use of higher-order representations. To alleviate these issues, here we take inspiration from random walk-based graph neural networks on graphs (Tönshoff et al., 2023) and explore the use of a random walk-based learning architecture for simplicial complexes. The advantages of this strategy are particularly apparent for simplicial complexes and related architectures. First, by choosing the number of random walks sampled, we can effectively trade off the computational demands with the expressivity of our architecture. Second, as we can compute fast 1D convolutions on the generated random walk trajectories via fast Fourier transforms, this saves us from computing far more expensive convolutional filters on simplicial complexes. Interestingly, it was shown in (Tönshoff et al., 2023) that there exist graphs that cannot be distinguished by the classical Weisfeiler-Leman (WL) test, that can be distinguished using such a random walk-based learning strategy, and vice versa that certain graphs cannot be distinguished that are distinguishable by the 1-WL test. This incomparability to message-passing schemes extends to

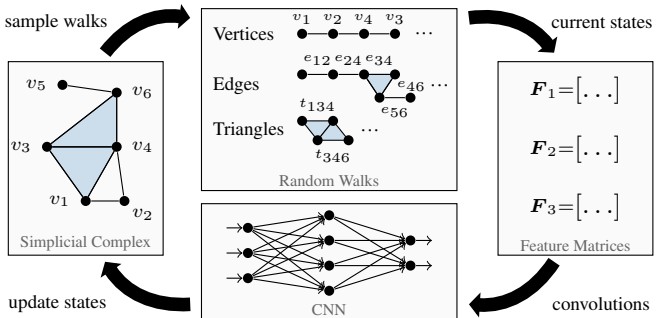

Figure 1: **Sketch of SCRaWl illustrating the individual steps of the model.** We sample a collection of random walks on the simplicial complex and transform them into walk feature matrices. These are then processed by a 1D convolutional network and pooled into updated simplex states.

the case of simplicial complexes, and as a result, our learning architecture has an expressivity that is not comparable to message-passing-based schemes such as (Bodnar et al., 2021b).

**Contribution**  We contribute SCRaWl, a novel neural network architecture for simplicial complexes that utilizes random walks on higher-order simplices to incorporate higher-order relations between entities into the neural network. This leads to an architecture different from existing simplicial neural networks. We prove that SCRaWl's expressiveness is incomparable to that of message-passing simplicial networks and show that our model outperforms existing approaches on real-world datasets.

**Related work**  Compared to corresponding graph learning problems, learning from data supported on simplicial complexes has so far received far less attention. Ebli et al. (2020) presented a basic convolutional neural network (SNN) for simplicial complexes, which was later extended by Yang et al. (2022a) to incorporate information flows between different orders of simplices (SCNN). Roddenberry et al. (2021) proposed a convolutional architecture for trajectory predictions that is rooted in tools from algebraic topology and derived three desirable properties for simplicial neural architectures: permutation equivariance, orientation equivariance, and simplicial awareness.

Bodnar et al. (2021b) proposed a message-passing simplicial network (MPSN), which adapts the proven message-passing concept from graph neural networks to simplicial complexes using messages to adjacent, upper-adjacent, and lower-adjacent simplices. They later extended this line of work to the domain of cell complexes (Bodnar et al., 2021a). Similarly, Goh et al. (2022) and Giusti et al. (2022) independently adapted the well-known attention mechanism from graph attention networks and constructed the simplicial attention networks SAT and SAN, respectively. Apart from this line of research focused on extending graph models to simplicial complexes, Keros et al. (2022) proposed a graph convolutional model for learning functions parametrized by the $k$-homological features of simplicial complexes.

Random walks on simplicial complexes have been considered in (Schaub et al., 2020; Alev & Lau, 2020; Parzanchevski & Rosenthal, 2017; Mukherjee & Steenbergen, 2016). Billings et al. (2019) and Hacker (2020) proposed two similar random walk-based representation approaches (simplex2vec and $k$-simplex2vec) to learn embeddings from co-appearing simplices in the walks, with the idea that co-appearing simplices should have similar embeddings while other simplices should be further apart in the embedding space. Yang et al. (2022b) extended this work with sc2vec, which takes more connections into account and therefore yields more effective embeddings. These unsupervised representation learning methods compute embeddings for simplices or simplicial complexes, which can then be used for downstream tasks such as classification, while SCRaWl is a supervised learning method that is trained end-to-end.

For graphs, several neural networks based on random walks have been proposed. Nikolentzos & Vazirgiannis (2020) proposed RWNN, a model that integrates a random walk-based kernel into a GNN architecture. Jin et al. (2022) and Eliasof et al. (2022) proposed two architectures that aggregate node features along random walks to learn information based on walk distances. Our work here extends the approach by Tönshoff et al. (2023), which proposed a random walk-based graph neural network

architecture that enriches the random walks with local structural information and processes the resulting feature matrices with a 1D convolutional neural network. Zhou et al. (2023) connect some of these concepts and extend ideas behind RWNN to simplicial complexes. In contrast to SCRaWl, they use walk probabilities to enrich the node and edge features for following message-passing steps, while we make use of explicit random walks as the main building block of our architecture.

Another popular abstraction tool for higher-order data is hypergraphs, for which several learning architectures have been proposed. In a similar vein to simplicial neural networks, graph neural networks have been abstracted to hypergraphs, e.g., convolutional networks (Yadati et al., 2019), attention networks (Chen et al., 2020), and message-passing networks (Heydari & Livi, 2022; Chien et al., 2022). Very recently, Behrouz et al. (2023) presented a walk-based hypergraph neural network that uses temporal walks to extract higher-order causal patterns. Compared to their approach, SCRaWl can capture more structural properties as part of the feature matrices resulting from the random walks.

**Outline**   The remainder of this manuscript is structured as follows. Section 2 gives an overview of the mathematical concepts of simplicial complexes and random walks on them. In Section 3, we present our proposed architecture SCRaWl in detail. Section 4 analyzes the expressiveness of SCRaWl compared to MPSN and Section 5 evaluates SCRaWl empirically on a variety of datasets.

## 2   Background

Here, we present an elementary overview of concepts used to process signals defined on simplicial complexes. For more details, see (Bredon, 1993; Hatcher, 2002) for background in algebraic topology and (Schaub et al., 2021; Roddenberry et al., 2022) for topological signal processing.

We use $[n]$ to denote the set of integers from $0$ to $n - 1$. Vectors are denoted by boldface lowercase letters $\boldsymbol{v}$, matrices and tensors by capital letters $\boldsymbol{F}$. For ease of notation, we index rows and columns from 0. $\boldsymbol{F}_{i,-}$ and $\boldsymbol{F}_{-,j}$ denote the $i$-th row and $j$-th column, respectively, of a matrix $\boldsymbol{F}$. $\boldsymbol{0}_n$ and $\boldsymbol{1}_n$ denote an $n$-dimensional all-zero and all-one vector, respectively.

An abstract *simplicial complex* (SC) (Bredon, 1993; Hatcher, 2002) $X$ consists of a finite set of points $\mathcal{V}$, and a set of non-empty subsets of $\mathcal{V}$ that is closed under taking non-trivial subsets. A $k$-*simplex* $\mathcal{S}^k \in X$ is a subset of $\mathcal{V}$ with $k + 1$ points and if $\mathcal{S}^k \in X$, then for all non-empty $\mathcal{S}^{k-1} \subset \mathcal{S}^k$, $\mathcal{S}^{k-1} \in X$. We denote the set of $k$-simplices in $X$ by $X_k$ and their cardinality by $n_k = |X_k|$. Furthermore, for $k > 0$ the *faces* $\mathfrak{F}(\mathcal{S}^k) = \left\{ \mathcal{S}^{k-1} \in X_{k-1} \mid \mathcal{S}^{k-1} \subset \mathcal{S}^k \right\}$ of $\mathcal{S}^k$ are the subsets of $\mathcal{S}^k$ with cardinality $k$. We set $\mathfrak{F}(\mathcal{S}^k) = \emptyset$ for $k = 0$. If $\mathcal{S}^{k-1}$ is a face of $\mathcal{S}^k$, $\mathcal{S}^k$ is called a *coface* of $\mathcal{S}^{k-1}$. We denote the set of cofaces of $\mathcal{S}^k$ by $\mathfrak{C}(\mathcal{S}^k)$.

We respectively denote the set of lower and upper adjacent simplices of $\mathcal{S}^k$ by:

$$N^{\downarrow}(\mathcal{S}^k) = \left\{ \mathcal{S}'^k \in X_k \mid \mathfrak{F}(\mathcal{S}^k) \cap \mathfrak{F}(\mathcal{S}'^k) \neq \emptyset \right\}, \tag{1}$$

$$N^{\uparrow}(\mathcal{S}^k) = \left\{ \mathcal{S}'^k \in X_k \mid \mathfrak{C}(\mathcal{S}^k) \cap \mathfrak{C}(\mathcal{S}'^k) \neq \emptyset \right\}. \tag{2}$$

Note that SCs can be understood as an extension of graphs: $X_0$ is the set of vertices, $X_1$ is the set of edges, $X_2$ is the set of filled-in triangles (not all 3-cliques have to be filled-in triangles), and so on.

Simplices are equipped with features $f_k : X_k \to \mathbb{R}^{d_k}$. For flexibility, we allow the feature spaces to be different for each simplex order, as is common in real-world data sets. If no features are supported on a simplex order, we set $d_k = 0$.

## 3   Method

At its core, SCRaWl extracts information from data supported on a simplicial complex by sampling random walks on the underlying complex. These walks are then transformed into feature matrices, which are used to update the states of the simplices in the subsequent steps. More precisely, SCRaWl consists of three steps, as are illustrated in Figure 1: First, we sample random walks on simplicial complexes, as described in Section 3.1. Second, as discussed in Section 3.2, we transform these walks into feature matrices that encode the random walks, the simplex states that appear on them, and the local adjacencies between simplices along the walk. Finally, we process these feature matrices with a simple convolutional neural network to update the hidden states of the simplices (see Section 3.3).

## 3.1 RANDOM WALKS ON SIMPLICIAL COMPLEXES

A walk of length $\ell$ on the $k$-simplices of a simplicial complex $X$ is a sequence of $k$-simplices $(v_0, \ldots, v_{\ell-1}) \in X_k^l$ such that $v_i$ and $v_{i+1}$ share a common face or coface $e_i$ for all $i \in [\ell]$, i.e., $v_{i+1} \in N^\downarrow(v_i) \cup N^\uparrow(v_i)$. Note that different from the case of a graph $e_i$ is not uniquely defined given $v_i$ and $v_{i+1}$, since $v_{i+1}$ can be reached by either a common face or a common coface. We keep track of the connection used to reach the next neighbor and define $W = (v_0, e_0, v_1, e_1, \ldots, e_{\ell-2}, v_{\ell-1})$.

While there are many possible sampling schemes for random walks on SCs, for simplicity, we only consider two elementary sampling methods: (a) uniform connection sampling, and (b) uniform neighbor sampling. For both sampling strategies, we obtain a random walk on the $k$-simplices of $X$ as follows: First, sample a random starting $k$-simplex $v_0 \sim \mathcal{U}(X_k)$ and then sample the subsequent simplices in a walk as described in the following subsections. Using this strategy we sample $m$ random walks on the simplicial complex, which can be chosen at runtime and can vary between training and prediction. In the case $m = |X|$, we opt to start a random walk from each simplex, i.e., we do not sample $v_0$ uniformly at random but choose every simplex once as $v_0$. Note that by choosing a smaller $m$ we can reduce the computational cost we incur for learning and processing the data.

**Uniform Connection Sampling** Starting from simplex $v_0$ we iteratively sample a face or coface $e_i \sim \mathcal{U}\big(\mathfrak{F}(v_i) \cup \mathfrak{C}(v_i)\big)$ and a neighbor

$$v_{i+1} \sim \begin{cases} \mathcal{U}\big(\mathfrak{C}(e_i)\big) & \text{if } e_i \in \mathfrak{F}(v_i) \\ \mathcal{U}\big(\mathfrak{F}(e_i)\big) & \text{if } e_i \in \mathfrak{C}(v_i) \end{cases} \tag{3}$$

of $v_i$ that is connected to $v_i$ via $e_i$ uniformly at random. This way, a neighbor with more connections to the current simplex is more likely to be sampled.

**Uniform Neighbor Sampling** Starting from simplex $v_0$ we iteratively sample a neighboring simplex $v_{i+1} \sim \mathcal{U}\big(N^\downarrow(v_i) \cup N^\uparrow(v_i)\big)$ and a suitable connection

$$e_i \sim \mathcal{U}\Big(\big(\mathfrak{F}(v_i) \cap \mathfrak{F}(v_{i+1})\big) \cup \big(\mathfrak{C}(v_i) \cap \mathfrak{C}(v_{i+1})\big)\Big) \tag{4}$$

uniformly at random. This way, higher-order simplices do not have a disproportionate influence on the walk due to their many low-order connections.

## 3.2 WALK FEATURE MATRICES

Based on the sampled collection of random walks $\{W_j\}_{j \in [m]}$ from the previous step and a local window size $s$, we transform each walk $W_j$ into feature matrix $\boldsymbol{F}_{W_j}$. The feature matrix consists of 6 sub-matrices: (a) one matrix for the features of the current simplex, (b) one matrix for the features of the face used to traverse to the current simplex, (c) one matrix for the features of the coface used to traverse to the current simplex, and (d) the last three matrices encode local structural information about the simplicial complex seen during the walk.

More formally, given a walk $W_j$ of length $\ell$ on $k$-simplices, embedding functions $f_{k'} : X_{k'} \to \mathbb{R}^{d_{k'}}$ for $i \in \{k-1, k, k+1\}$, and a local window size $s > 0$, we define the walk feature matrix $\boldsymbol{F}_{W_j}$ as

$$\boldsymbol{F}_{W_j} = \begin{bmatrix} \boldsymbol{F}^\rightarrow_{W_j} & \boldsymbol{F}^\downarrow_{W_j} & \boldsymbol{F}^\uparrow_{W_j} & \boldsymbol{I}_{W_j,s} & \boldsymbol{A}^\downarrow_{W_j,s} & \boldsymbol{A}^\uparrow_{W_j,s} \end{bmatrix} \in \mathbb{R}^{\ell \times (d_k + d_{k-1} + d_{k+1} + 3s - 2)}. \tag{5}$$

The first sub-matrix $\boldsymbol{F}_{W_j} \in \mathbb{R}^{\ell \times d_k}$ records the features of the $k$-simplices appearing on the walk $W$:

$$(\boldsymbol{F}^\rightarrow_W)_{i,-} = f(v_i) \text{ for } i \in [\ell]. \tag{6}$$

The second and third sub-matrices $\boldsymbol{F}^\downarrow_{W_j}$ and $\boldsymbol{F}^\uparrow_{W_j}$ record the features of the faces and cofaces, respectively, that have been used in the random walk to get from the last simplex to the current one:

$$\left(\boldsymbol{F}^\downarrow_{W_j}\right)_{i,-} = \begin{cases} f(e_{i-1}), & \text{if } i > 0 \text{ and } |e_{i-1}| < k \\ \boldsymbol{0}_{d_{k-1}}, & \text{otherwise.} \end{cases} \tag{7}$$

$$\left(\boldsymbol{F}^\uparrow_{W_j}\right)_{i,-} = \begin{cases} f(e_{i-1}), & \text{if } i > 0 \text{ and } |e_{i-1}| > k \\ \boldsymbol{0}_{d_{k+1}}, & \text{otherwise.} \end{cases} \tag{8}$$

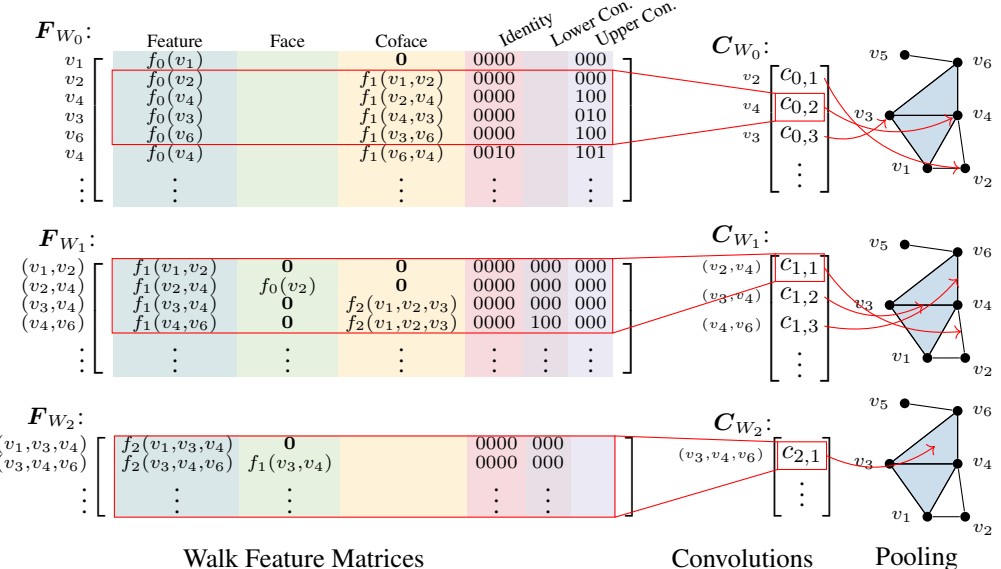

Figure 2: **Information flow in SCRaWl modules for simplex orders** 0, 1, **and** 2. Using the walks shown in Figure 1, we compute the feature matrices $F_0$, $F_1$, and $F_2$ based on simplex features $f$ and a window size of $s = 4$. The feature matrices are convolved with a 1D CNN while keeping track of the central simplex within the window (values to the left of the convolution matrices). Convolved matrices are pooled into simplices grouped by the central simplices.

Note that whenever a face or a coface feature is present, the entries of the other matrix are zero. Further, assume that $d_{-1} = d_{k_{\max}+1} = 0$ and hence $\boldsymbol{F}^{\downarrow}_{W_j}$ and $\boldsymbol{F}^{\uparrow}_{W_j}$ are empty for walks on 0-simplices and on maximum order simplices, respectively, where $k_{\max}$ is the maximum order in the SC.

The remaining three sub-matrices $\boldsymbol{I}_{W_j} \in \{0, 1\}^{\ell \times s}$, $\boldsymbol{A}^{\downarrow}_{W_j} \in \{0, 1\}^{\ell \times (s-1)}$ and $\boldsymbol{A}^{\uparrow}_{W_j} \in \{0, 1\}^{\ell \times (s-1)}$ record structural information in the form of local identity and local adjacency features:

$$(\boldsymbol{I}_{W_j,s})_{i,i'} = \begin{cases} 1, & \text{if } i \geq i' \text{ and } v_i = v_{i'} \\ 0, & \text{otherwise.} \end{cases} \tag{9}$$

$$(\boldsymbol{A}^{\downarrow}_{W_j,s})_{i,i'} = \begin{cases} 1, & \text{if } i > i' \text{ and } v_{i'-i} \in N^{\downarrow}(v_i) \\ 0, & \text{otherwise.} \end{cases} \tag{10}$$

$$(\boldsymbol{A}^{\uparrow}_{W_j,s})_{i,i'} = \begin{cases} 1, & \text{if } i > i' \text{ and } v_{i'-i} \in N^{\uparrow}(v_i) \\ 0, & \text{otherwise.} \end{cases} \tag{11}$$

In other words, $\boldsymbol{I}_{W_j,s}$ keeps track of whether the current simplex has been visited before within the local window $s$, while $\boldsymbol{A}^{\downarrow}_{W_j,s}$ and $\boldsymbol{A}^{\uparrow}_{W_j,s}$ keep track of whether the current simplex is lower or upper adjacent to a previously visited simplex within the local window $s$, respectively.

Figure 2 (left) shows an example of the walk feature matrices for the walks on the vertices, the edges, and the triangles shown in Figure 1. Note in particular the last three sub-matrices for each walk: For $W_0$, vertex $v_4$ is visited twice, once in the third and once in the fifth step, thus we have $(\boldsymbol{I}_{W_0,4})_{5,2} = 1$. In the same walk, we go from $v_1$ via the edges $(v_1, v_2)$ and $(v_2, v_4)$ to $v_4$, which is also directly connected with $v_1$. Thus we have $(\boldsymbol{A}^{\uparrow}_{W_0,4})_{2,0} = 1$. Similarly, for $W_1$, we go from $(v_2, v_4)$ over two steps to $(v_4, v_6)$, which share a common vertex $v_4$. Thus we have $(\boldsymbol{A}^{\downarrow}_{W_1,4})_{3,0} = 1$.

### 3.3 SCRaWl Module

A SCRaWl module on layer $t$ operating on order $k$-simplices takes as input the random walks on $k$-simplices and the current hidden states $\boldsymbol{H}^{t-1}_k$, $\boldsymbol{H}^{t-1}_{k-1}$, and $\boldsymbol{H}^{t-1}_{k+1}$ of the previous layer $t-1$. These

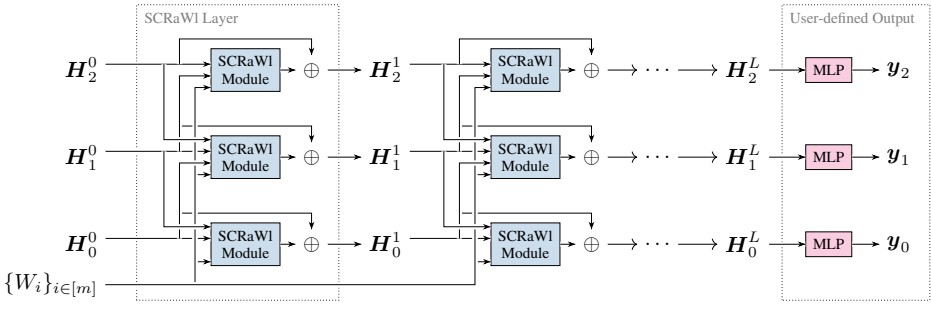

Figure 3: **Architecture of a SCRaWl model operating on simplex orders** $0$, $1$, **and** $2$ **with** $L$ **SCRaWl layers and an user-defined output layer.** We reuse the same collection of random walks $\{W_j\}_{j \in [m]}$ in each SCRaWl module for performance reasons.

matrices are then used in lieu of the simplex, face, and coface features in the walk feature matrices, respectively. The module first computes the feature matrices $\boldsymbol{F}_{W_j}$ for each walk $W_j$ as described in the previous subsection and stacks them into a tensor $\boldsymbol{F}_k$.

We process the walk features $\boldsymbol{F}_k$ by a 1D convolutional network $\mathrm{CNN}_k^t$ (without padding) performed over the walk steps to obtain the convolved matrix $\boldsymbol{C}_k^t \in \mathbb{R}^{m_k \times (l-s) \times d}$, where $m_k$ is the number of walks on $k$-simplices. The CNNs can be configured flexibly depending on the task.

For each row of the convolved matrix $\boldsymbol{C}_k^t$, we separately keep track of the simplex in the center row of the convolution window, as is illustrated in Figure 2 with the indices next to the convolution matrices. For each $k$-simplex, we pool all entries of $\boldsymbol{C}_k^t$ that correspond to that simplex using mean or sum pooling. This operation results in a vector $\boldsymbol{p}_k^t(v)$ for each $k$-simplex $v$, which is then fed into a trainable multilayer perceptron $\mathrm{Update}_k^t$ to obtain the updated hidden state $\boldsymbol{H}_k^t(v)$.

The process of a SCRaWl module is illustrated in Figure 2. As can be seen, the new hidden state of $k$-simplices depends on the hidden state of upper and lower adjacent simplices, and hence, over multiple layers, information is propagated through different simplex orders.

### 3.4 COMPLETE SCRAWL ARCHITECTURE

Based on the aforementioned building blocks, we can assemble the SCRaWl model, as illustrated in Figure 3: Given a range of simplex orders $[K]$ and the number of layers $L$ as hyperparameters, the model consists of $L$ SCRaWl layers, each consisting of $K$ SCRaWl modules, one for each simplex order. Within one layer, SCRaWl modules run in parallel for each simplex order $k \in [K]$. We add skip connections for each module. Information flows between different simplex orders in between each layer in the form of the hidden states of faces and cofaces appearing in the random walks.

At the end of the last layer, we obtain hidden states $\boldsymbol{H}_k^L$ for each simplex order $k \in [K]$. These can then be processed by user-defined output layers, e.g., by adding multilayer perceptrons that transform the hidden states into a classification or regression output, to obtain the final output of the model.

### 3.5 PERFORMANCE CONSIDERATIONS

Most of the computational complexity of SCRaWl is due to the sampling of random walks. Hence, we only sample the random walks once in each epoch and reuse this collection in all layers, i.e., each module transforms them into module-specific feature matrices with different feature values (first three sub-matrices) but reuses the same random walks for that. This is also depicted in Figure 3 with the same collection $\{W_j\}_{j \in [m]}$ of random walks passed to each layer. We further describe how to sample the random walks more efficiently using boundary maps.

The structure of a simplicial complex can be encoded by boundary maps $\boldsymbol{B}_k$, which record the incidence relations between $(k-1)$-simplices and $k$-simplices (Hatcher, 2002). Rows and columns of $\boldsymbol{B}_k$ are indexed by $(k-1)$-simplices and by $k$-simplices, respectively. The entry $(i,j)$ is set to 1 or 0 depending on whether the $i$-th $(k-1)$-simplex is incident to the $j$-th $k$-simplex. Thus, the

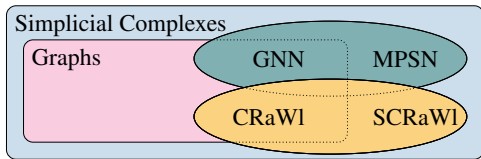

Figure 4: **Expressiveness relations of SCRaWl to other neural network models.** SCRaWl and MPSN are strict extensions of their graph counterparts CRaWl and message-passing GNNs, respectively, and their expressive power is incomparable.

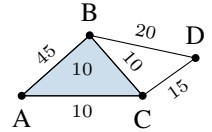

| Paper | Authors | Citations |
|---|---|---|
| 1 | A, B | 35 |
| 2 | B, D | 20 |
| 3 | C, D | 15 |
| 4 | A, B, C | 10 |

Figure 5: **Example of a citation simplicial complex.** Each vertex represents an author and connections (edges and triangle) indicate co-authorship. The value of the connection is the total number of citations of the papers the authors have written together.

matrix $B_1$ is the unsigned vertex-to-edge incidence matrix, and $B_2$ is the edge-to-triangle incidence matrix. For ease of notation, we define $B_0 = B_{k+1}0$. Further define $S_k = \text{hstack}(B_k^\top, B_{k+1})$.

For uniform connection sampling on the $k$-simplices, we can efficiently sample the next connection $j$ from $S_k$, as the connections for the $i$-th $k$-simplex are precisely those with value one in $(B_k)_{-,i}$ and $(B_{k+1})_{i,-}$. If the sampled index $j \leq n_{k-1}$, the walk transitions over the face $j$, otherwise over the coface $j - n_{k-1}$. The next walk simplex can then be sampled uniformly from $S_k^\top$.

For uniform neighbor sampling, compute $A_k = B_k^\top B_k + B_{k+1}B_{k+1}^\top$ and $\hat{A}_k = \text{sign}(A_k)$. The neighbor of the $i$-th $k$-simplex can be sampled uniformly at random from $(\hat{A}_k)_{i,-}$.

## 4 EXPRESSIVE POWER

It is known that the expressive power of message-passing GNNs with anonymous inputs is limited by the WL graph isomorphism test (Morris et al., 2019; Xu et al., 2019; Morris et al., 2023) and thus such GNNs cannot distinguish between non-isomorphic graphs that cannot be separated by color refinement. In (Bodnar et al., 2021b, Definition 5) a simplicial version of the WL test (SWL) was introduced and it was shown that the expressiveness of message-passing simplicial networks (MPSN) is limited by this test. The following theorem asserts that for SCRaWl this analysis does not apply.

**Theorem 1.** *The expressiveness of SCRaWl is incomparable to the expressiveness of an MPSN, i.e., there are simplicial complexes that can be distinguished by SCRaWl but not by MPSN and vice versa.*

*Proof.* Note that when applied to 1-simplicial complexes and considering vertex colorings, SWL is equivalent to WL and by extension, the expressiveness of MPSNs corresponds to the expressiveness of GNNs (Bodnar et al., 2021b). Note also that SCRaWl is equivalent to CRaWl when applied to 1-simplicial complexes. Thus the theorem follows from (Tönshoff et al., 2023, Theorem 1). □

The expressiveness inclusions of the various architectures are shown in Figure 4. This implies that the expressiveness of SCRaWl is incomparable to the WL test and its simplicial counterpart.

## 5 EXPERIMENTAL RESULTS

We evaluate SCRaWl on a variety of datasets and compare it to other simplicial neural networks[1]. Following the training procedure of Dwivedi et al. (2023), we use the Adam optimizer with an initial learning rate $10^{-3}$. The learning rate decays with a factor of $0.5$ if the validation loss does not improve for 10 epochs. Training is stopped once the learning rate drops below $10^{-6}$. For a full list of hyperparameters and a more detailed description of the training setup, see Appendix A.

### 5.1 SEMANTIC SCHOLAR CO-AUTHORSHIP NETWORK

Following Ebli et al. (2020), we use SCRaWl to impute missing citation counts for a subset of the *Semantic Scholar co-authorship* network. Vertices represent distinct authors, and a paper with $k + 1$

---

[1]Source code and datasets are available at `https://git.rwth-aachen.de/netsci/scrawl`.

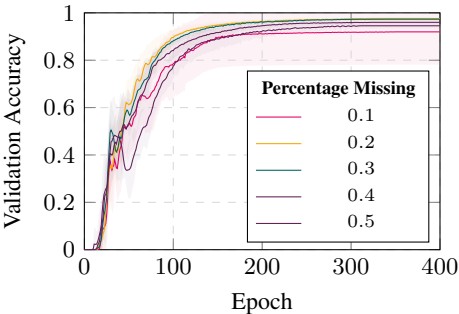 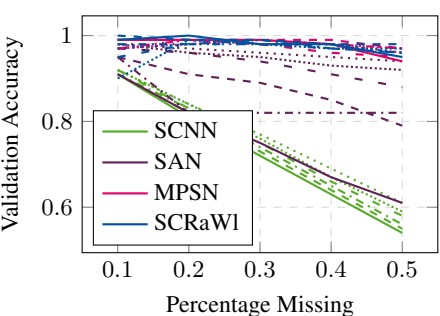

Figure 6: **Imputation accuracies on the Semantic Scholar dataset with different percentages of missing citations.** We report the mean and standard deviation over 10 runs. Left: Training progress of SCRaWl. Right: Accuracies of SCRaWl, SIN (MPSN), SAN, and SCNN for each simplex order as more and more data is missing. Increasingly lighter lines indicate higher simplex orders.

authors implies a $k$-simplex connecting these authors in the network (see Figure 5). The value on a $k$-simplex is the sum of citations of the papers written together by the respective $k + 1$ authors (including papers with more than these authors). Citation values are randomly deleted at a rate of 10%, 20%, 30%, 40%, and 50% and replaced by the median of the remaining known citations.

We report the imputation accuracies of SCRaWl in Figure 6 (left) and compare them with $k$-simplex2vec (Hacker, 2020), SNN (Ebli et al., 2020), SCNN (Yang et al., 2022a), SAT (Goh et al., 2022), SAN (Giusti et al., 2022), and the MPSN architecture presented in (Bodnar et al., 2021b). For readability, we omitted the results for $k$-simplex2vec, SNN, and SAT in Figure 6 (right). Full results are available in Table 1 in the appendix. Following the previous work, a citation value is correctly imputed if the prediction is within $\pm 5\%$[2] of the true value. We repeat each experiment 10 times and report the mean and standard deviation for missing rates of 10%, 20%, 30%, 40%, and 50%. The left plot shows the overall accuracies of our model on the different missing rates while the right plot compares the accuracies of the different models on individual simplex orders.

We see that SCRaWl reaches accuracies of 95% to 99% for all missing rates after about 175 epochs. Across all experiments, we consistently see a drop in accuracy around epoch 30 to 50, which is then overcome. We forced a minimum of 100 epochs to avoid early stoppings within this drop. We also see that SCRaWl is on par with MPSN and outperforms all other models for most missing rates and simplex orders. Especially for missing rates above 10%, the improvement is substantial. In addition, imputation accuracies are more consistent across different simplex orders for SCRaWl, whereas, for the other models, the accuracy declines faster for smaller simplices than for higher-order simplices.

## 5.2 SOCIAL CONTACT NETWORKS

In a second set of experiments, we perform vertex classification on the *primary-school* and *high-school* social contact datasets (Stehlé et al., 2011; Chodrow et al., 2021). The datasets are constructed based on students' interactions recorded by wearable sensors. Vertices represent students, and a simplex corresponds to a group of students that were in proximity to each other at a given time. Vertices are labeled with the classroom to which the student belongs, resulting in a total of 12 classes for the primary school dataset and 10 classes for the high school dataset.

We use a cross-entropy loss to train the network on the dataset with 40% of the vertex classes missing and report the validation accuracy of the imputed classes in Figure 7 (left). For the primary school dataset, SCRaWl reaches an average accuracy of $0.927 \pm 0.026$ after about 160 epochs, and for the high school dataset a perfect accuracy of $1.0 \pm 0.0$ after about 150 epochs.

We compare this result with the performance of MPSN on the same datasets (middle). Although MPSN generally converges faster in our experiment, it only achieves an average accuracy of $0.727 \pm 0.033$ and $0.943 \pm 0.033$ for the primary and high school dataset, respectively, and is thus outperformed by SCRaWl, especially on the primary school dataset. Comparisons with SCNN,

---

[2] The SNN paper used a threshold of $\pm 10\%$ which was later changed to $\pm 5\%$.

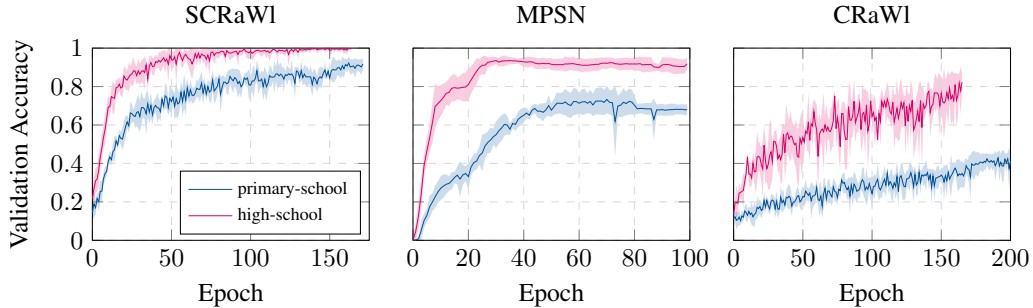

Figure 7: **Vertex classification accuracies of SCRaWl, MPSN, and CRaWl on the social contact datasets.** We report the mean and standard deviation of the validation accuracies over 5 runs.

$k$-simplex2vec and some hypergraph models can be found in Table 2 in the appendix. The other simplicial approaches did not perform well on the datasets, with SCNN reaching an average accuracy of $0.166 \pm 0.056$ for the primary school dataset and $0.228 \pm 0.075$ for the high school dataset. $k$-simplex2vec reached an average accuracy of $0.398 \pm 0.011$ and $0.308 \pm 0.019$ for the primary and high school dataset, respectively. The hypergraph approaches performed consistently at most around $0.9$ accuracy on the high school dataset, $10$ percent below our result. For the primary school dataset, CAt-Walk marginally outperformed SCRaWl with an average accuracy of $0.932 \pm 0.025$.

**Importance of Higher-Order Interactions**    We repeated the same experiment using the graph-based CRaWl architecture on the underlying graph skeleton of the datasets, i.e., the graph only records pairwise proximities between students but does not record whether interactions between more than two people occurred. Keeping all other parameters the same, we found that CRaWl performance is clearly worse than SCRaWl on both datasets (see Figure 7). This highlights the importance of higher-order interactions in the dataset and the ability of SCRaWl to capture them.

## 6   CONCLUSION

We proposed a simplicial neural network architecture SCRaWl based on random walks and 1D convolutions. The architecture builds on the idea of (Tönshoff et al., 2023) to use sampled random walks for graph learning tasks. We demonstrated that these ideas can be extended to work on simplicial complexes if we carefully adjust the notion of random walks. We showed that the resulting architecture outperforms existing simplicial neural network architectures on a co-authorship network.

**Other Complexes**    While we focused on simplicial complexes in this work, the architecture can naturally be applied to other types of complexes such as (regular) cell complexes without any changes. Informally, a cell complex is a collection of cells, where each cell's boundary is also in the complex. Starting with points as 0-cells, 1-cells as lines with their two endpoints as their boundary, a $k$-cell is defined such that its boundary consists of a collection of $(k-1)$-cells. A polygon for instance is a 2-cell with a boundary consisting of a set of lines. This highlights the difference to a simplicial complex, as a 2-simplex must form a triangle, while a 2-cell may be any polygon.

**Future Work**    The empirical results demonstrate the potential of the proposed architecture. However, there are several avenues for future work worth exploring. First, we plan to investigate the performance of SCRaWl in a broader set of applications. Simplicial complexes have gained a lot of interest for edge flow problems, and it remains to be seen how well SCRaWl performs on these tasks. While we have paid special attention to the computational complexity in our implementation, due to the many simplices and hence walk feature matrices in a simplicial complex compared to a graph, we expect that using SCRaWl with one random walk per simplex will not scale sufficiently. Approximation guarantees for different sampling schemes are thus an interesting direction for future work.

ACKNOWLEDGMENTS

We acknowledge funding by the Ministry of Culture and Science (MKW) of the German State of North Rhine-Westphalia ("NRW Rückkehrprogramm") and the European Union (ERC, HIGH-HOPeS, 101039827). Views and opinions expressed are however those of the authors only and do not necessarily reflect those of the European Union or the European Research Council Executive Agency. Neither the European Union nor the granting authority can be held responsible for them.

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

Table 1: **Imputation accuracies on the Semantic Scholar dataset with varying percentages of missing citations.** We report the mean and standard deviation of the validation accuracies over 10 runs. We compare SCRaWl to the simplicial neural networks SNN, SCNN, SAT, SAN, and MPSN as well as to the representation learning algorithm $k$-simplex2vec.

| Validation Size | Method | Simplex Order | | | | | |
| --- | --- | --- | --- | --- | --- | --- | --- |
| | | 0 $n_0 = 352$ | 1 $n_1 = 1474$ | 2 $n_2 = 3285$ | 3 $n_3 = 5019$ | 4 $n_4 = 5559$ | 5 $n_5 = 4547$ |
| 10% | $k$-simplex2vec | $0.02 \pm 0.014$ | $0.07 \pm 0.013$ | $0.40 \pm 0.040$ | $0.78 \pm 0.021$ | $0.96 \pm 0.007$ | $0.93 \pm 0.069$ |
| | SNN | $0.91 \pm 0.003$ | $0.91 \pm 0.002$ | $0.90 \pm 0.004$ | $0.91 \pm 0.004$ | $0.90 \pm 0.016$ | $0.90 \pm 0.008$ |
| | SCNN | $0.91 \pm 0.004$ | $0.91 \pm 0.002$ | $0.91 \pm 0.002$ | $0.92 \pm 0.001$ | $0.92 \pm 0.002$ | $0.92 \pm 0.002$ |
| | SAT | $0.18 \pm 0.000$ | $0.31 \pm 0.000$ | $0.28 \pm 0.001$ | $0.34 \pm 0.001$ | $0.53 \pm 0.001$ | $0.55 \pm 0.001$ |
| | SAN | $0.91 \pm 0.004$ | $0.95 \pm 0.019$ | $0.95 \pm 0.019$ | $\mathbf{0.97} \pm 0.016$ | $\mathbf{0.98} \pm 0.009$ | $\mathbf{0.98} \pm 0.007$ |
| | SIN (MPSN) | $\mathbf{0.99} \pm 0.004$ | $\mathbf{0.99} \pm 0.003$ | $\mathbf{0.99} \pm 0.002$ | $\mathbf{0.97} \pm 0.008$ | $\mathbf{0.97} \pm 0.004$ | $\mathbf{0.97} \pm 0.003$ |
| | **SCRaWl** | $\mathbf{0.99} \pm 0.035$ | $\mathbf{1.00} \pm 0.009$ | $0.98 \pm 0.020$ | $0.95 \pm 0.138$ | $0.90 \pm 0.252$ | $\mathbf{0.97} \pm 0.020$ |
| 20% | $k$-simplex2vec | $0.03 \pm 0.023$ | $0.08 \pm 0.013$ | $0.46 \pm 0.014$ | $0.81 \pm 0.021$ | $0.95 \pm 0.009$ | $0.92 \pm 0.079$ |
| | SNN | $0.81 \pm 0.006$ | $0.82 \pm 0.003$ | $0.82 \pm 0.005$ | $0.83 \pm 0.004$ | $0.82 \pm 0.012$ | $0.83 \pm 0.007$ |
| | SCNN | $0.81 \pm 0.007$ | $0.82 \pm 0.003$ | $0.83 \pm 0.003$ | $0.83 \pm 0.002$ | $0.84 \pm 0.002$ | $0.84 \pm 0.002$ |
| | SAT | $0.18 \pm 0.000$ | $0.20 \pm 0.000$ | $0.29 \pm 0.001$ | $0.35 \pm 0.001$ | $0.50 \pm 0.001$ | $0.58 \pm 0.001$ |
| | SAN | $0.82 \pm 0.008$ | $0.91 \pm 0.024$ | $0.82 \pm 0.008$ | $0.96 \pm 0.004$ | $0.96 \pm 0.013$ | $0.97 \pm 0.009$ |
| | SIN (MPSN) | $\mathbf{0.99} \pm 0.005$ | $\mathbf{0.99} \pm 0.002$ | $\mathbf{0.99} \pm 0.004$ | $\mathbf{0.99} \pm 0.008$ | $\mathbf{0.98} \pm 0.006$ | $\mathbf{0.99} \pm 0.004$ |
| | **SCRaWl** | $\mathbf{1.00} \pm 0.006$ | $\mathbf{0.99} \pm 0.006$ | $0.98 \pm 0.015$ | $\mathbf{0.98} \pm 0.019$ | $\mathbf{0.99} \pm 0.008$ | $\mathbf{0.98} \pm 0.013$ |
| 30% | $k$-simplex2vec | $0.01 \pm 0.008$ | $0.07 \pm 0.003$ | $0.41 \pm 0.019$ | $0.79 \pm 0.020$ | $0.95 \pm 0.006$ | $0.92 \pm 0.088$ |
| | SNN | $0.72 \pm 0.006$ | $0.73 \pm 0.004$ | $0.73 \pm 0.005$ | $0.75 \pm 0.002$ | $0.75 \pm 0.002$ | $0.75 \pm 0.003$ |
| | SCNN | $0.72 \pm 0.005$ | $0.73 \pm 0.004$ | $0.74 \pm 0.003$ | $0.75 \pm 0.002$ | $0.76 \pm 0.002$ | $0.77 \pm 0.002$ |
| | SAT | $0.19 \pm 0.000$ | $0.33 \pm 0.001$ | $0.25 \pm 0.001$ | $0.33 \pm 0.000$ | $0.47 \pm 0.001$ | $0.53 \pm 0.001$ |
| | SAN | $0.75 \pm 0.021$ | $0.89 \pm 0.021$ | $0.82 \pm 0.008$ | $0.94 \pm 0.004$ | $0.95 \pm 0.005$ | $0.96 \pm 0.005$ |
| | SIN (MPSN) | $\mathbf{0.99} \pm 0.004$ | $\mathbf{0.99} \pm 0.002$ | $0.98 \pm 0.004$ | $\mathbf{0.98} \pm 0.009$ | $\mathbf{0.99} \pm 0.001$ | $\mathbf{0.98} \pm 0.001$ |
| | **SCRaWl** | $0.98 \pm 0.015$ | $\mathbf{0.99} \pm 0.004$ | $\mathbf{0.99} \pm 0.003$ | $\mathbf{0.98} \pm 0.016$ | $\mathbf{0.98} \pm 0.012$ | $\mathbf{0.98} \pm 0.010$ |
| 40% | $k$-simplex2vec | $0.01 \pm 0.003$ | $0.08 \pm 0.007$ | $0.46 \pm 0.024$ | $0.77 \pm 0.016$ | $0.92 \pm 0.008$ | $0.90 \pm 0.085$ |
| | SNN | $0.63 \pm 0.007$ | $0.64 \pm 0.003$ | $0.65 \pm 0.003$ | $0.66 \pm 0.004$ | $0.67 \pm 0.009$ | $0.67 \pm 0.008$ |
| | SCNN | $0.63 \pm 0.006$ | $0.64 \pm 0.003$ | $0.65 \pm 0.002$ | $0.66 \pm 0.002$ | $0.67 \pm 0.003$ | $0.69 \pm 0.002$ |
| | SAT | $0.20 \pm 0.000$ | $0.29 \pm 0.000$ | $0.22 \pm 0.000$ | $0.43 \pm 0.001$ | $0.51 \pm 0.001$ | $0.50 \pm 0.001$ |
| | SAN | $0.67 \pm 0.019$ | $0.85 \pm 0.028$ | $0.82 \pm 0.008$ | $0.91 \pm 0.009$ | $0.93 \pm 0.011$ | $0.95 \pm 0.016$ |
| | SIN (MPSN) | $\mathbf{0.98} \pm 0.002$ | $\mathbf{0.99} \pm 0.001$ | $\mathbf{0.98} \pm 0.003$ | $0.96 \pm 0.007$ | $\mathbf{0.98} \pm 0.002$ | $\mathbf{0.98} \pm 0.008$ |
| | **SCRaWl** | $\mathbf{0.98} \pm 0.009$ | $0.98 \pm 0.009$ | $0.97 \pm 0.021$ | $\mathbf{0.97} \pm 0.028$ | $0.97 \pm 0.012$ | $\mathbf{0.98} \pm 0.013$ |
| 50% | $k$-simplex2vec | $0.07 \pm 0.015$ | $0.07 \pm 0.008$ | $0.36 \pm 0.016$ | $0.80 \pm 0.022$ | $0.91 \pm 0.007$ | $0.95 \pm 0.003$ |
| | SNN | $0.54 \pm 0.007$ | $0.55 \pm 0.005$ | $0.56 \pm 0.003$ | $0.57 \pm 0.003$ | $0.59 \pm 0.004$ | $0.60 \pm 0.005$ |
| | SCNN | $0.54 \pm 0.006$ | $0.55 \pm 0.004$ | $0.56 \pm 0.003$ | $0.58 \pm 0.003$ | $0.59 \pm 0.003$ | $0.61 \pm 0.002$ |
| | SAT | $0.19 \pm 0.000$ | $0.30 \pm 0.001$ | $0.22 \pm 0.000$ | $0.32 \pm 0.001$ | $0.43 \pm 0.000$ | $0.48 \pm 0.001$ |
| | SAN | $0.61 \pm 0.019$ | $0.79 \pm 0.043$ | $0.82 \pm 0.008$ | $0.88 \pm 0.015$ | $0.92 \pm 0.007$ | $0.94 \pm 0.011$ |
| | SIN (MPSN) | $0.94 \pm 0.004$ | $0.97 \pm 0.006$ | $\mathbf{0.97} \pm 0.008$ | $0.95 \pm 0.002$ | $0.96 \pm 0.004$ | $\mathbf{0.97} \pm 0.009$ |
| | **SCRaWl** | $\mathbf{0.95} \pm 0.029$ | $\mathbf{0.98} \pm 0.009$ | $0.96 \pm 0.009$ | $\mathbf{0.96} \pm 0.032$ | $\mathbf{0.97} \pm 0.014$ | $0.96 \pm 0.035$ |

SNN and SCNN have been reported by Yang et al. (2022a), SAT and SAN have been reported by Giusti et al. (2022).

## A   TRAINING DETAILS AND RESULTS

For all experiments, we use the Adam optimizer with an initial learning rate of $10^{-3}$. The learning rate is reduced by a factor of $0.5$ if the validation loss does not improve for 10 epochs. Training is stopped once the learning rate drops below $10^{-6}$.

The walk length $\ell$ is the primary hyperparameter that determines the receptive field of the model. As the citation counts in the Semantic Scholar dataset can be imputed well with only local information, we choose a walk length of $5$ for this dataset. On other datasets, the model is trained with a walk length of $50$. The walk length can be increased independently in prediction, but we did not make use of this option in our experiments.

For the Semantic Scholar dataset, we ran SCRaWl modules on simplex orders $k \in \{0, \ldots, 5\}$, i.e., simplices of order 6 appear only as static coface features in the random walks and simplices of order 7 do not influence the model. Using uniform connection sampling, we computed $m = \sum_{i=0}^{n} n_i$ random walks, i.e., one random walk for each simplex of order $k \in \{0, \ldots, 5\}$. While SCRaWl modules can be configured individually for each simplex order and each layer, we found that the same configuration for all modules works well for this dataset: Each module is configured with a local window size of $s = 4$, a kernel size of $d_{\text{kern}} = 8$, a hidden feature size of $d = 32$, and a mean pooling operation. Complete results for the semantic scholar experiments are given in Table 1.

For the social contact datasets, we use 4 layers and run SCRaWl modules on simplex orders $k \in \{0, \ldots, 3\}$. We compute one random walk for each simplex. Each SCRaWl module is again configured identically with a local window size of $s = 8$, a kernel size of $d_{\text{kern}} = 8$, a hidden feature size of $d = 128$, and a mean pooling operation. The final vertex embeddings are fed into a 2-layer MLP with ReLU activation. Complete results for the social contact experiments are given in Table 2.

Table 2: **Vertex classification accuracies on the social contact datasets.**

|  |  | Dataset | |
| --- | --- | --- | --- |
|  |  | primary-school | high-school |
| Dataset Details | $n_0$ | 242 | 327 |
|  | $n_1$ | 8317 | 5818 |
|  | $n_2$ | 5139 | 2370 |
|  | $n_3$ | 381 | 238 |
|  | Target Classes | 12 | 10 |
|  | Random Baseline | 0.107 | 0.135 |
| Simplicial Methods | $k$-simplex2vec[a] | $0.398 \pm 0.011$ | $0.308 \pm 0.019$ |
|  | SCNN | $0.166 \pm 0.056$ | $0.228 \pm 0.075$ |
|  | SIN (MPSN) | $0.727 \pm 0.043$ | $\mathbf{0.943 \pm 0.033}$ |
|  | CRaWl | $0.415 \pm 0.030$ | $0.823 \pm 0.042$ |
|  | **SCRaWl** | $\mathbf{0.927 \pm 0.026}$ | $\mathbf{1.000 \pm 0.000}$ |
| Hypergraph Methods[b] | HyperGCN | $0.852 \pm 0.031$ | $0.849 \pm 0.036$ |
|  | AllSetTransformer | $0.898 \pm 0.026$ | $0.908 \pm 0.031$ |
|  | CAt-Walk | $\mathbf{0.932 \pm 0.025}$ | $0.907 \pm 0.050$ |

[a] For vertex classification, $k$-simplex2vec is oblivious to any higher-order connection beyond edges.
[b] Results on hypergraphs have been reported by Behrouz et al. (2023).

Table 3: **List of hyperparameters and training setup used for each experiment.**

| Parameter | Semantic Scholar | Social Contacts |
| --- | --- | --- |
| Walk Length $\ell$ | 5 | 50 |
| Sampling Method | uniform connection | uniform connection |
| Number of Layers $L$ | 3 | 4 |
| Max. Simplex Order $K$ | 5 | - |
| Local Window Size $s$ | 4 | 8 |
| Kernel Size $d_{\text{kern}}$ | 8 | 8 |
| Hidden Feature Size $d$ | 32 | 32 |
| Pooling | mean | mean |
| Total number of Trainable Parameters | 171k | 132k |
| Optimizer | Adam; LR $= 10^{-3}$ | |
| LR Scheduler | Reduce on plateau: factor 0.5; patience 10 | |
| Stopping Criterion | LR $< 10^{-6}$ | |

A full list of hyperparameters used for each experiment is given in Table 3. The model has been implemented using PyTorch (Paszke et al., 2019) and TopoX (Hajij et al., 2024). We adapted the code from Tönshoff et al. (2023) to implement SCRaWl.

We used Bodnar et al. (2021b) reference implementation for MPSN and conducted the experiments using the author's default parameters with 4 layers, 128 hidden features, ReLU nonlinearities, and sum aggregations. Experiments for SCNN were conducted using the implementation provided by TopoX with 3 layers, 64 hidden features, and ReLU nonlinearities. We used our own implementation of $k$-simplex2vec.

## B   ABLATION STUDY

In this ablation study, we investigate the influence of different hyperparameters and other aspects on the performance of SCRaWl. For all experiments, the training setup and hyperparameters not under

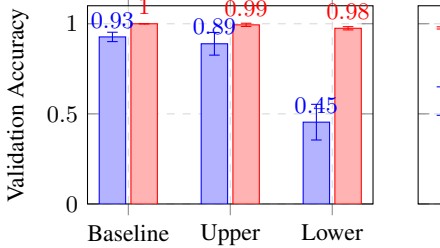 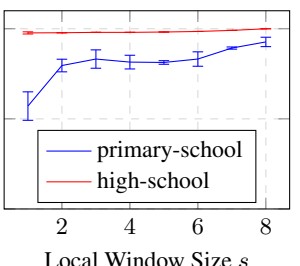 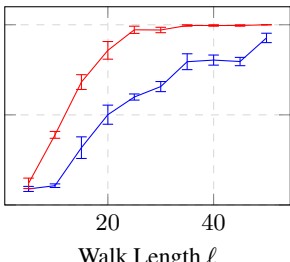

Figure 8: **Ablation study on the social contact datasets.** Left: Influence of upper and lower adjacency on the performance of SCRaWl. Middle: Influence of the local window size $s$. Right: Influence of the walk length $\ell$. We report the mean and standard deviation of the validation accuracies over 5 runs.

investigation are set to the same values as in the main experiments. Figure 8 shows the results of the ablation study on the social contact datasets.

**Lower vs. Upper Adjacency**   During a random walk, the next simplex can either be reached by traversing over a face or a coface of the current simplex. We are interested in the impact of either form of adjacency, i.e., we compare the performance of SCRaWl when using only lower adjacency or only upper adjacency. For simplices that don't have a face or don't have a coface, respectively, the random walk stays at the current simplex. We make the exception that for node-level walks, upper adjacency over edges is always allowed, as otherwise, no meaningful walks would be possible.

For the primary school dataset, we see that only using upper adjacency yields a significantly better performance of $0.889 \pm 0.063$ than only using lower adjacency with $0.454 \pm 0.099$. In this case, only using upper adjacency is thus only marginally below the baseline performance of $0.927 \pm 0.026$. On the easier-to-learn high school dataset, the difference between using only upper or lower adjacency is smaller with $0.994 \pm 0.009$ vs. $0.975 \pm 0.009$, compared to the baseline with $1.0 \pm 0.0$.

These results support our claim that — at least for these datasets — higher-order interactions give additional insight into the structure of the dataset and methods that capture these (higher-order) structures can perform better.

**Influence of Local Window Size** $s$   The local window $s$ is a parameter that influences which structural features of the input simplicial complex are visible to the model. We thus expect that larger windows improve the prediction performance while also increasing the computational complexity.

We validate this hypothesis by training SCRaWl on the social contact datasets with different local window sizes $s \in \{1, \ldots, 8\}$. As expected, we see that the prediction performance of SCRaWl improves with larger windows: For the primary school dataset, the performance increases almost monotonically from $0.571 \pm 0.079$ with $s = 1$ to $0.927 \pm 0.026$ with $s = 8$. For the easier-to-learn high school dataset the performance is only marginally worse with $0.977 \pm 0.006$ for $s = 1$ vs. $1.0 \pm 0.0$ for $s = 8$.

**Influence of Walk Length** $\ell$   Similar to the local window size $s$, we expect that larger walk lengths $\ell$ positively influence the prediction performance of SCRaWl, as longer walks ensure that important structural features of the simplicial complex are captured and processed by the model.

To that end, we trained SCRaWl on the social contact datasets with different walk lengths $\ell \in \{5, 10, \ldots, 50\}$. The prediction performance again improved almost monotonically with increasing walk lengths for both datasets, with the high school dataset reaching accuracies above $0.98$ for $\ell \geq 25$ already. For the primary school dataset, the performance reached its best accuracy of $0.927 \pm 0.026$ only for $\ell = 50$.

We note that the choice of walk length $\ell$ and window size $s$ has a considerable impact on the computational cost of the model and its value should be chosen carefully to balance predictive performance and computational complexity.

