# OpenReview forum: "Learning From Simplicial Data Based on Random Walks and 1D Convolutions"
_ICLR.cc/2024/Conference — ICLR 2024 poster_

### Official Review · Reviewer_vWSA · 2023-10-23

**Soundness:** 4 excellent
**Presentation:** 4 excellent
**Contribution:** 4 excellent
**Rating:** 8
**Confidence:** 5

**Summary:**

This paper presents a novel algorithm for learning on simplicial
complexes, i.e. generalisations of graphs. As opposed to relying on the
message passing paradigm, the paper uses random walks (or rather,
*sampled* random walks) to featurise the input data, subsequently
employing 1D convolutions as a type of local update mechanism.
Experiments on several data sets demonstrate the efficacy of the method.

**Strengths:**

The paper is exceptionally well written and describes highly-original
research. Given the recent interest in extending graph learning to
higher-order structures, the work is also timely and highly-relevant.

Moreover, I appreciate that this paper presents alternatives to the
prevailing message passing paradigm. As amply demonstrated in the
manuscript, this leads to a different class of expressivity, thus
opening the door to further explorations.

The current write-up is sufficiently detailed to be reproducible, and
all concepts are explained sufficiently well. This paper was truly
a pleasure to read and review.

**Weaknesses:**

Only minor weaknesses cropped up, which can be easily addressed in
a revision (please also see my questions below).

- The impact of parameter choices could be studied more carefully.

- A discussion on the utility of a simplicial perspective might enrich
  the paper. As it stands now, the experiments deal with simplicial
  data. The scope of the paper could be improved substantially if the
  simplicial perspective was shown to be *crucial* for good performance.
  I want to stress that I consider this to be a minor weakness since the
  paper as such already makes a strong contribution under the assumption
  that simplicial complexes are the 'right' thing to model the data.
  Showing this empirically would just be a cherry on top of this cake.

  (the experiment on social contact networks is a good start for this,
  but I would appreciate a more in-depth discussion **if possible**)

- Unless I am mistaken, the proper verb form should be 'convolved' as
  a opposed to 'convoluted' (in Figure 2).

- As a minor style issue: please use `\citep` and `\citet` (when using
  `natbib`) consistently. The former is meant for parenthetical
  citations, the latter for in-text citations.

- There are some inconsistencies (mostly capitalisation / venues) in the
  references.

- When it comes to explaining expressivity in the context of WL,
  a [recent survey](https://arxiv.org/abs/2112.09992) might be useful as
  an additional reference. (The related work covered at the moment is
  sufficient; this is just a suggestion. The review form unfortunately
  lacks a field for 'additional comments')

**Questions:**

1. How are the random walk parameters (length / number of walks)
   affecting performance? Could you show predictive performance as
   a function of these parameters (or, even better, provide theoretical
   guarantees)?

2. Would the proposed method also extend to *hypergraphs* or other
   combinatorial complexes? This seems to be the case, given that merely
   a notion of a random walk is required.

3. Given the performance considerations, are there any practical
   limitations in terms of data set size that could be given? This might
   be useful to assess the suitability for certain data sets in advance.

---

> ### Author Response · Authors · 2023-11-22
>
> Thank you for your detailed response. Let us address each point separately:
>
> **Parameter Choice**
>
> We added an ablation study to the paper that explores the influence of considered adjacency, the size of the local window, and the length of the random walks on the performance on the social contact datasets.
>
> **Utility of Simplicial Perspective**
>
> We agree that highlighting the importance a simplicial perspective on datasets would be benefitial.
> However due to space constraints we considered this outside the scope of our paper.
> The comparison between SCRaWl and CRaWl (Figure 7) provides an initial indication towards this topic and other authors considered this comparison as well, e.g. Bodnar et al, 2021.
>
> We argue that this topic should be dealt with in a more dedicated work and on a more abstract level not focused on one particular neural network architecture but rather with different graph neural networks and simplicial neural networks in mind.
>
> **Application to other Complexes**
>
> The approach should also work on combinatorial complexes, provided an analogous notion of a random walk is defined.
> This is similar to the case of cell complexes as explained in the conclusion.
> For hypergraphs, the notion of upper and lower adjacency is not well defined, which is why we did not consider them in our work.
> Of course, ignoring this (e.g., by merging the respective columns into one) would yield a valid approach, but we did not investigate the performance of such a method in detail here.
>
> **Dataset Size**
>
> For larger datasets, reducing the number of random walks is a viable option to reduce complexity.
> Choosing one random walk per simplex makes is very likely to cover each simplex sufficiently often as a center simplex in a local window, but this property is neither ensured nor necessary.
> It is also worth considering to use smaller parameters during training and larger parameters during inference.
>
> **Minor Things**
>
> We added the suggested survey paper as reference for expressivity in context with WL and fixed the minor style issues.

---

> > ### Comment · Reviewer_vWSA · 2023-11-23
> >
> > Thanks a lot—I'm very excited about your work and believe it will have substantial impact!

---

### Official Review · Reviewer_Wmrd · 2023-10-31

**Soundness:** 2 fair
**Presentation:** 3 good
**Contribution:** 2 fair
**Rating:** 5
**Confidence:** 4

**Summary:**

The paper presents SCRaWI, a neural network model to encode simplicial complexes based on higher-order random walks on simplices. SCRaWI samples several higher-order random walks followed by fast 1D convolutions to learn the structural properties of the Simplicial Data.

**Strengths:**

The paper is well-written and easy to follow. Also, it is well-organized with enough illustrative figures. The proposed method is simple, yet potentially powerful, which makes it feasible for large and real-world cases. More, specifically, the model design allows choosing the number of sampled random walks as a hyperparameter, which provides a trade-off between computational cost and the expressivity of the model. Also, using fast Fourier transforms to perform convolutions on sampled walks is an interesting idea, which potentially can improve efficiency and training time.

**Weaknesses:**

The paper has missed some important related studies or detailed discussions about them. As an example, there are several random walk-based methods [1, 2] in a simple graph, which uses a similar idea. While they have been briefly mentioned, there are some important questions about them that are not discussed in the paper. Is there any connection between SCRaWl and these methods? Or is SCRaWl the extended version of these studies to higher-order data? A very similar study on higher-order graphs is CATWALK [3], which is a hypergraph learning method that similarly uses higher-order random walks to learn higher-order patterns. I think, given the fact that hypergraphs can be seen as a general case of simplicial complexes, their walk sampling is exactly the same as this work, which limits the novelty of this paper's idea and its model design.

The proposed method is simple, which is a desirable property **if** the effectiveness of the method is theoretically or experimentally shown. That is, the novelty of the model architecture is limited, so it is expected that extensive experimental or theoretical results will support the method's performance. However, in the paper, the experimental study is very limited, e.g.:

1. There are only three datasets from two different domains. Please note that co-authorship networks and social networks usually have similar properties so it would be much better if the authors could provide more datasets with different domains, specifically in drug-drug or chemical networks (NDC, and/or NDC Substances), and communication networks (Email Enron).

2. There is a lack of ablation study on the method architecture. Accordingly, it is not clear what is the contribution of each component of SCRaWl. How using a simple random walk instead of a higher-order random walk can affect the performance? How does each of the six features contribute to the performance of SCRaWl?

3. The main motivations and claims in the paper are not supported by experiments. For example, when the main motivation is to address the inefficiency of the existing methods, I think it is needed to see how SCRaWl performs compared to existing methods in terms of time and how proposed components are improving the efficiency.

4. Hypergraphs are another paradigm to represent higher-order interactions. There is a lack of comparison with state-of-the-art hypergraph learning methods [3, 4, 5].



In addition to the above points, as discussed in the "Introduction" section, there is a trade-off between computational cost and the power of SCRaWl. It would be great if you could show this trade-off in the experiments as well.


  $$ $$


[1] Random walk graph neural networks. NeurIPS 2020.

[2] Walking out of the Weisfeiler Leman hierarchy: Graph learning beyond message passing. TMLR 2023.

[3] CAT-Walk: Inductive Hypergraph Learning via Set Walks. NeurIPS 2023.

[4] Teasing out missing reactions in genome-scale metabolic networks through hypergraph learning. Nature Communications 2023.

[5] You are allset: A multiset function framework for hypergraph neural networks. ICLR 2022.



$ $

$ $


---
---
``Post Rebuttal``
---
---
---

I thank the authors for their response.

My concerns remain unaddressed in the rebuttal phase. Specifically,

* There is a lack of discussion with very similar works (e.g., [2]). As mentioned in my initial review, the existing discussion, which briefly mentioned that ``here we take inspiration from random walk-based graph neural networks on graphs (Tönshoff et al., 2023)`` is not enough. To my understanding, the idea of using 1D Convolutions is previously discussed in [2], and the only remaining contribution in this paper is to design higher-order random walks on simplicial data. The proposed walk also is similar to the walk that is previously discussed in [3]. The only difference is that the random walk in [3] is temporal, which can be simply adjusted and so be restricted to a single timestamp. The authors claim that [3] has been published after the submission deadline, while as far as I understood [3] has been available from Jun 2023 (almost 3-4 months before the deadline).

* I understand that the novelty is subjective and coming up with a model that combines existing studies but provides new insights is significant. However, I couldn’t find enough experimental evaluations to support the paper, and my request to perform experiments on more datasets remains unaddressed. More specifically, **(1)** The current experimental evaluation is limited to only three datasets (two of which come from the same domain). There are several benchmark datasets that are publicly available and are suitable for the experimental settings used in this paper. **The authors didn’t respond to this comment.**  **(2)** There is a lack of comparison with some important baselines (e.g., [3, 5]). The authors reported the results provided in [3], but it is not clear to me that the experimental setting used in [3] is the same as this paper. Is the hyperparameter tunning procedure for all the methods the same? Is the preprocessing step on the datasets the same? I think reporting the results from another work is unfair due to different experimental setups, data splitting, etc.  **(3)** The provided results in the appendix show the sensitivity of the method to hyperparameters. The authors' response did not address the concern about the lack of an ablation study. Specifically, my two questions in the initial review remain unaddressed: (i) How can using a simple random walk instead of a higher-order random walk affect performance? (ii) How does each of the features contribute to the performance of SCRaWl? (iii) What is the contribution of 1D Convolutions? What would happen if we use a simple linear layer?

* As I mentioned in the initial review, the main motivations and claims in the paper are not supported by experiments. **The authors didn’t respond to this comment.** More specifically, when the main motivation is to address the inefficiency of the existing methods, I think it is necessary to see how SCRaWl performs compared to existing methods in terms of time and how proposed components are improving efficiency.


---
$$ $$

**After the rebuttal phase and also discussion with other reviewers, I raised my score to 5.**

$$ $$

**Questions:**

Please see the Weaknesses. In summary, my suggestions are here: Please add

1. more discussions about existing methods [1, 2]. Also please discuss [3] and its differences with your method.
2. more datasets, different experimental settings, and hypergraph learning-based baselines.
3. ablation studies to show the contribution of each component.
4. scalability and efficiency evaluation.

---

> ### Author Response · Authors · 2023-11-22
>
> **References**
>
> We incorporated the suggested references into our work.
> References [1] and [2] have already been cited in our related work section before.
> In particular, [2] is the primary work we built upon by extending their approach to simplicial complexes.
> We believe this is made clear in the current version of the paper already, e.g. in the introduction: "[...] here we take inspiration from random walk-based graph neural networks on graphs (Tönshoff et al., 2023) and explore the use of a random walk-based learning architecture for simplicial complexes."
>
> It is our understanding that [4] focuses on hyperedge link prediction and their proposed method is therefore not (directly) applicable to the tasks considered in our paper.
> Similarly, [3] deals with temporal data, which is not the focus of our paper, though they also explain how their approach can be applied to node classification on static hypergraphs.
> We however want to point out that this work was published after the submission deadline of our paper.
>
> We added a paragraph to our related works section that mentions hypergraphs as another paradigm to represent higher-order interactions and reference to some noticeable hypergraph neural networks, including the proposed references [3] and [5].
> We further extended the comparison on the social contact datasets with hypergraph neural networks (Table 3 in the appendix).
>
> **Ablation Study**
>
> We added an ablation study on the social contact datasets to the appendix where we compare the influence of the window size $s$, the walk length $\ell$, and of lower and upper adjacency when sampling the walks.

---

### Official Review · Reviewer_WBPx · 2023-11-01

**Soundness:** 3 good
**Presentation:** 3 good
**Contribution:** 3 good
**Rating:** 8
**Confidence:** 4

**Summary:**

The paper proposes a method called SCRAWL which explores simplicial complexes-based learning representations for graph datasets using random walks and 1D convolution on simplices. This work is built on top of the recent work CRAWL which uses random walks and 1D conv directly on graphs. The authors performed two tasks vertex classification and missing citation counts on two benchmark datasets.

**Strengths:**

1. The paper is well presented and motivated.
2. Additional information from simplices seems to improve embeddings. The results are interesting.
3. An ablation study is provided to show the robustness and compare it against existing works.
4. The expressivity is equivalent to CRAWL. So, the theorem by CRAWL holds for this too.

**Weaknesses:**

1. The major concern I could see is memory and computation as it contains the matrices for k levels of simplices. And, each walk consists of six matrices.
2. It is not clear how long the walk is and how many walks are computed for each of the k-simplices. Assuming m is for the collection of all walks from all k-simplices.
3. Why are walks passed in every layer? Is it not enough to use it only at the input?
4. How are the output from k-simplices combined for the final output? I could see three different outputs in Fig. 3.
5. Fig 6. results are on par with MPSN. Justification of why it works differently on two different kinds of datasets would be much appreciated. Results are improved significantly for social networks although it.

**Questions:**

Why are walks computed on the fly or they are just sampled on the fly? Can’t we precompute the walks which is just one time processing before training?

---

> ### Author Response · Authors · 2023-11-22
>
> 1. We agree that the memory footprint is considerable, especially for large datasets.
>    However, this is generally the case for simplicial complexes as the simplicial property forces exponentially many sub-simplices for any higher order simplex.
>    The feature matrices themselves can be computed on the fly and are very sparse in the later columns (containing the structural information about connectivity and identity).
>    The first columns contain features that any reasonable approach needs to store as well.
>    An advantage of our approach is, however, that we can vary the scalability, e.g., by selecting the window sizes and walk lengths, which is not possible for classical message passing architectures. Investigating such trade-offs in more detail (more broadly: how to optimally leverage higher-order information, while keeping computations tractable) is an interesting topic we intend to explore in future work in more detail.
> 2. In our experiments, we sampled one random walk per simplex.
>    Walk lengths are set to 5 for the semantic scholar dataset and 50 for the other datasets.
>    These parameters are mention in the text of section 5 and Table 1 in the appendix.
>    These parameters are of course hyperparameters and can be tuned for the respective dataset.
> 3. The feature matrices change in every layer (because the hidden states are updated), thus the random walks are passed to every layer.
> 4. This depends on the dataset at hand and the considered targets.
>    For the semantic scholar dataset, the outputs $y_0, ..., y_5$ are compared as is with the targets.
>    For the social contact dataset, only the outputs $y_0$ are considered.
> 5. For the semantic scholar dataset, the results of SCRaWl and MPSN are close to perfect, so we cannot expect any strong improvement beyond MPSN.
>    The social contact datasets show that SCRaWl is able to outperform MPSN.
>
> Regarding your questions:
> Yes, the walks can be precomputed as well. This is actually also implemented in our code.
> However, this does not have any influence on imputation accuracy, but only on training time, which is why we did not include it in the paper.
> Notice that the same random walks should not be reused in every epoch to avoid overfitting to these specific walks.

---

> > ### Comment · Reviewer_WBPx · 2023-11-23
> > **Thanks**
> >
> > Thanks for the feedback. I have read the responses to my concerns. Most of the concerns are addressed. I will consolidate everything in the final comments and rating.

---

### Official Review · Reviewer_6GSS · 2023-11-04

**Soundness:** 2 fair
**Presentation:** 3 good
**Contribution:** 2 fair
**Rating:** 5
**Confidence:** 2

**Summary:**

This paper proposes a new learning method on simplicial complexes based on random walks and 1-D convolution encodings.

**Strengths:**

1. The idea of using random walks to tackle the high complexity of learning on simplicial complexes is a natural idea. The overall design also looks reasonable to me.
2. Experiments show that the proposed method outperforms some of the most recent methods.

**Weaknesses:**

1. Compared to this work, a recent work [1] seems to provide more principled insights into exactly the same topic. Can the authors discuss more of its edge over this work?

2. The experiments are conducted on a relatively limited number of datasets. I would hope to see more datasets from diverse domains being used.

[1] Facilitating Graph Neural Networks with Random Walk on Simplicial Complexes, NeurIPS 2023

**Questions:**

See Weaknesses.

---

> ### Comment · Reviewer_6GSS · 2023-11-22
>
> The rebuttal period is approaching the end. Can the authors please respond to my review? Also, I have read Reviewer Wmrd's review, and would like to echo with their concern about comparison with CATWALK and lack of experiment on more diverse dataset. Can the authors please also address these points?

---

> ### Author Response · Authors · 2023-11-22
>
> We thank the reviewer for their feedback.
>
> We would like to point out that the mentioned paper was published after the submission deadline of this paper.
> However, we agree that the paper provides interesting results and discuss it now in our work.
> Notice that the paper is similar to RWNN (already in our related work) in that the authors employ random walk probabilities as additional node and edge features, which enhance the main graph neural network.
> Different to RWNN, they use random walks on simplicial complexes (e.g. by lifting).
> In contrast, in our work, the concrete random walks are not used as enhancement, but as the main building block of the network.

---

### Meta-Review · Area_Chair_oXLT · 2023-12-04

**Metareview:**

The paper introduces SCRAWL, an innovative approach for learning representations in graph datasets through simplicial complexes. Building on the previous CRAWL framework, SCRAWL utilizes random walks and 1D convolution on simplices instead of directly on graphs. Departing from the traditional message passing paradigm, the method employs sampled random walks to featurize input data and employs 1D convolutions as a local update mechanism. Experimental results on benchmark datasets for vertex classification and missing citation counts showcase the effectiveness of the proposed method, highlighting its potential for learning on simplicial complexes.

This paper provides an interesting method and some reviewers are excited about the results. In contrast, some concerns exist; the difference from the CAT-WALK method should be further clarified in discussion and experiments. This paper is on a borderline.

**Justification For Why Not Higher Score:**

This is a borderline paper.

**Justification For Why Not Lower Score:**

N/A

---

### Decision · Program_Chairs · 2024-01-16

Accept (poster)